# Efficacy of the Aqueous Extract of *Azadirachta indica* Against the Marine Parasitic Leech and Its Phytochemical Profiling

**DOI:** 10.3390/molecules26071908

**Published:** 2021-03-29

**Authors:** Balu Alagar Venmathi Maran, Dawglas Josmeh, Jen Kit Tan, Yoong Soon Yong, Muhammad Dawood Shah

**Affiliations:** 1Borneo Marine Research Institute, Universiti Malaysia Sabah, Kota Kinabalu, Jalan UMS 88450, Malaysia; wizards.xj@gmail.com; 2Department of Biochemistry, Universiti Kebangsaan Malaysia Medical Centre, Kuala Lumpur 56000, Malaysia; jenkittan@ukm.edu.my; 3Laboratory Centre, Xiamen University Malaysia, Jalan Sunsuria, Bandar Sunsuria, Sepang 43900, Malaysia; yoongsoon.yong@xmu.edu.my

**Keywords:** LC-Q Exactive HF Orbitrap MS, anti-parasitic, metabolites, aquaculture, grouper, control, *Zeylanicobdella arugamensis*

## Abstract

*Zeylanicobdella arugamensis* (Hirudinea)*,* a marine parasitic leech, not only resulted in the mortality of the host fish (Groupers) but also caused economic losses. The current study aimed to elucidate the antiparasitic efficacy of the aqueous extract of the *Azadirachta indica* leaves against *Z. arugamensis* and to profile the composition via LC-Q Exactive HF Orbitrap mass spectrometry. Different concentrations (25, 50 and 100 mg/mL) of *A. indica* extract were prepared and tested on the parasitic leeches. The total mortality of leeches was noticed with an exposure to the *A. indica* aqueous extract. The average times required for the aqueous extract at concentrations of 25, 50 and 100 mg/mL to kill the leeches were 42.65 ± 9.20, 11.69 ± 1.11 and 6.45 ± 0.45 min, respectively, in a dose-dependent manner. The Orbitrap mass spectrometry analysis indicated the presence of five flavonoids (myricetin 3-O-galactoside, trifolin, isorhamnetin, quercetin and kaempferol), four aromatics (4-methoxy benzaldehyde, scopoletin, indole-3-acrylic acid and 2,4-quinolinediol), three phenolics (p-coumaric acid, ferulic acid and phloretin) and two terpenoids (pulegone and caryophyllene oxide). Thus, our study indicates that *A. indica* aqueous extract is a good source of metabolites with the potential to act as a biocontrol agent against the marine parasitic leech in aquaculture.

## 1. Introduction 

In Malaysia, parasitic infestation is a serious problem for different types of fish species, and several parasites have been reported to be reared in open floating net-cages [1,2,3]. Aquacultured groupers showed a greater variety of parasites and higher intensities of infestation than wild groupers [2,4]. Some of the commonly found parasites in Malaysian cage aquacultures are monogeneans (*Benedenia* spp*., Neobenedenia* spp.), copepods (*Caligus epidemicus*, *Caligus* spp.) and piscicolids [1,3,4,5].

The infestation of the marine parasitic leech *Zeylanicobdella arugamensis* (Annelida: Hirudinea: Piscicolidae) (Figure 1A,B) spread rapidly in Southeast Asian countries [3,4,5,6]. In Malaysia, the marine leech was first reported in a grouper (*Epinephelus coioides*) reared in floating cages with a 0.4-percent prevalence [3]; later on, the leeches were frequently isolated from various major species of marine fish reared in cages such as hybrid groupers (*Epinephelus fuscoguttatus* × *E.lanceolatus*), groupers (*E. fuscoguttatus, E. lanceolatus*), snappers (*Lutjanus johnii*, *L. argentimaculatus* and *L. stellatus*) and sea bass (*Lates calcarifer*) [3,4,5]. Further, around 60 percent of moribund sea bass fingerlings nurtured in cages were found to be infested with *Z. arugamensis* [7]. The ectoparasitic leech also served as a vector for the transfer of pathogens and resulted in the mortality of the host fishes in a short period [8]. The control of leech infestation is vital for the management of the aquaculture industry. An expensive and harmful chemical, especially formalin, is used for the control of leech infestation, which is not conducive for an ecofriendly aquaculture policy [9]. The best alternative is the application of the natural product as a biocontrol agent due to the presence of various metabolites with less or zero toxicity [10].

Extracts and essential oils from a natural product can act as herbicide, antimicrobial, anticancer and antiparasitic agents [11,12,13,14]. *Azadirachta indica* (neem plant) belongs to the family Meliaceae, commonly found in the Indo-Malayan region and other parts of the world [15,16]. Various parts of the plant, such as fruits, seeds, leaves, bark and roots, are a good source of bioactive compounds with antimicrobial, antipyretic, anti-inflammatory and antiparasitic properties [15,16,17]. The antimicrobial activity of *A. indica* leaves’ solvent extracts against fish pathogenic bacteria *(Aeromonas veronii, Aeromonas hydrophila, Acinetobacter junii, Acinetobacter tandoii, Acinetobacter* spp. and *Pseudomonas stutzer*) isolated from Blackspot barb (*Dawkinsia filamentosa*) have been reported [18]. The aqueous extract of the plant leaf at a concentration of 150 mg/l has been reported to be effective against the pathogenic infection caused by *Citrobacter freundii* in tilapia [19]. The essential oils of *A. indica* have been reported to have an antimicrobial property against *Enterococcus faecalis, Aerococcus viridans, Pseudomonas aeruginosa, Proteus mirabilis* and *Escherichia coli* and an antiparasitic activity against *caligid* parasites on seabass [20,21]. The data have revealed that *A. indica* extracts are toxic to over 400 species of insect pests, some of which have developed resistance to chemically prepared pesticides [15,22]. The plant is also used for the removal of intestinal worms [15]. Currently, no data is available regarding the antiparasitic properties of *A. indica* aqueous extracts against the marine parasitic leech. Hence, this study aimed to elucidate the antiparasitic efficacy of the aqueous extract of *A. indica* against *Z. arugamensis* and to profile the composition using LC-Q Exactive HF Orbitrap mass spectrometry. 

## 2. Results

### 2.1. Physiochemical Parameters

The water quality parameters of the controls and plant-treated groups are provided in Table 1. Slight changes in the pH of the plant solutions were noticed when compared to the control groups, which could be due to the presence of metabolites with an acidic nature in the extract while the rest of the parameters remained constant.

### 2.2. Antiparasitic Properties of the Aqueous Extract of A. indica

The mortality time and percentage of the leeches treated with aqueous extracts of the plant were calculated (Table 2, Figure 1C–F). A total mortality of the leeches was noticed in negative control (Figure 1D) and all the plant-treated groups (Figure 1E–F), and the time taken was from 6.45 to 42.65 min. No mortality of the leeches was noticed in the normal control group (Figure 1C) until 720 min (12 h).

### 2.3. LC-Q Exactive HF Orbitrap Mass Spectrometry Analysis of the Aqueous Extract of A. indica

In the current study, a total of 42 compounds were identified using Q Exactive HF Orbitrap mass spectrometry (Table 3). Among these 42 compounds, there are five flavonoids (myricetin 3-O-galactoside, trifolin, isorhamnetin, quercetin and kaempferol), four aromatics (4-methoxy benzaldehyde, scopoletin, indole-3-acrylic acid and 2,4-quinolinediol), three phenolics (p-coumaric acid, ferulic acid and phloretin) and two terpenoids (pulegone and caryophyllene oxide). 

## 3. Discussion 

The marine leech *Z. arugamensis* is a notorious ectoparasite and distributed throughout the Indian Ocean [3,4,5,23,24]. For many years, fish farmers used toxic antiparasitic chemotherapeutics and insecticides to prevent or control parasitic infestations in aquaculture [25,26,27]. The accumulation of these chemical residues in water has caused impacts on the environment and may have lethal or sublethal effects on nontarget organisms [28]. For example, in Norway, when Neguvon and Nuvon pesticides were applied to control the copepod *Lepeophtheirus salmonis* (Caligidae) in salmon net-pen farming, there were harmful effects on several crustaceans near the farms [29]. However, plants are a good alternative and can be applied to control parasitic infestation as a natural remedy for sustainable aquaculture to avoid the negative effects of pesticides [10]. They show zero or less toxicity to the environment due to their biodegradability and are a great source of bioactive compounds [30,31]. 

In this study, we selected *A. indica* due to its antimicrobial, anti-inflammatory, antipyretic, insecticidal and acaricidal nature [15,16,17,22] and determined the antiparasitic potential of the aqueous extracts of *A. indica* leaves. The exposure of the aqueous extract resulted in the total mortality of leeches in a dose-dependent manner (Figure 1). In three different doses used, all leeches were killed in an average period of 6.45 ± 0.45 min (100 mg/mL), 11.69 ± 1.11 min (50 mg/mL) and 42.65 ± 9.20 min (25 mg/mL). Previous studies have shown the effect of plant extracts on different fish parasites. The methanol extract of *Dillenia suffruticosa* (Dilleniaceae, tropical shrub) was applied against *Z. arugamensis* at a concentration of 100 mg/mL, and it took 14.39 and 4.88 min to kill all the leeches [32]. However, our treatment using the aqueous extract of *A. indica* took less time than *D. suffruticosa* to kill all the leeches. Further, the exposure of the methanol extract and chromatographic fraction (fraction 3) of *Nephrolepis biserrata* (Nephrolepidaceae, perennial fern) was applied against *Z. arugamensis* at a concentration of 100 and 2.5 mg/mL and killed all the leeches in 4.88 and 1.92 min, respectively [33,34]. Similarly, the methanol extracts of *Allium sativum* (Amaryllidaceae, garlic) (600 μg/mL) were tested against the aquatic leech *Limnatis nilotica* (Hirudinidae) and took 144.55 min for total mortality [35]. The essential oil of *A. indica* was applied against a crustacean parasite *Caligus* (Caligidae) infestation on the Asian seabass *Lates calcarifer* (Latidae) and showed a 100% mortality of *Caligus* within 5760 min (96 h) at a rate of 10 ppm [21]. The extract of *Artemisia annua* (Asteraceae, aromatic herbaceous plant, Sweet Annie) was reported to be effective in 30 to 180 min against monogenean parasites of the cultured airbreathing catfish *Heterobranchus longifilis* at concentrations ranging from 50 to 200 mg/L [36]. Thus, in comparison to the above-mentioned plant, the aqueous extract of *A. indica* resulted in the total mortality of leeches in less than 7 min at a higher concentration. Furthermore, we explained the principle behind the antiparasitic nature of the plant via the identification of the different responsible metabolites using Q Exactive HF Orbitrap mass spectrometry. 

Among the compounds identified by Orbitrap mass spectrometry, some of them have been reported to have antiparasitic properties, especially on mammalian parasites, including quercetin, kaempferol, phloretin, trifolin, caryophyllene oxide and nicotinamide. The metabolites have also been reported in the plant extracts of *Trigonella foenum-graecum* [37], *Moringa oleifera* [38], Golden delicious apple pomace [39], *Zanthoxylum bungeanum* [40], *Pinus eldarica* [41] and *Oryza sativa* [42]. Some reports revealed that quercetin was effective against several mammalian parasites such as *Cryptosporidium parvum* (*Cryptosporidiidae*) and *Encephalitozoon intestinali* (Unikaryonidae), responsible for diarrheal diseases [43], *Leishmania amazonensis* (Trypanosomatidae), responsible for leishmaniasis [44], and *Trypanosoma* sp. (Trypanosomatidae), responsible for vector-borne disease and trypanosomiasis [45,46]. Quercetin directly induced apoptosis of *Trypanosoma brucei gambiense* without affecting the viability of the host cell [46]. Kaempferol inhibited the growth of the parasite *Entamoeba histolytica* (Entamoebidae), responsible for amoebiasis, by altering cytoskeleton proteins [47]. It is also well known for its antiproliferative [48], antidiabetic [49], antioxidative [50], anti-inflammatory [51] and anticancer properties [52]. Thus, we believe that the metabolites found in the aqueous extracts of *A. indica* could be responsible for the antiparasitic effect. Hence, we suggest that a natural-based treatment could be a viable alternative to chemicals and is effective for eco-friendly and sustainable aquaculture.

## 4. Materials and Methods

### 4.1. Chemicals

Formalin (37% aqueous formaldehyde solution) and sodium bicarbonates were obtained from Sigma, Leica, Microsystem, and Germany. Methanol (HPLC grade) was purchased from Merck (Darmstadt, Germany). LCMS-grade acetonitrile, water and formic acid were obtained from Fisher Scientific (Thermo Fisher Scientific, Waltham, MA, USA). Regenerated cellulose syringe filters with a 0.22-µm pore size and 13-mm diameter were obtained from Thermo Scientific (Thermo Fisher Scientific, Waltham, MA, USA). 

### 4.2. Plant collection

The leaves of the plant *A. indica* (Figure 2) were collected from Universiti Malaysia Sabah, Kota Kinabalu (5.7346° N, 115.9319° E), Sabah, East Malaysia. The identification of the plant was carried out at the Institute for Tropical Biology and Conservation, Universiti Malaysia Sabah, Kota Kinabalu.

### 4.3. Extraction

The leaves of the plant were rinsed with distilled water and dried in an oven at 37 °C. The dried plant was ground separately in a heavy-duty grinder. About 100 g of the dry plant powder was boiled with distilled water for 10 min with a 1:10 ratio (sample to the amount of distilled water) using a stirring hot plate. The decoctions were removed and allowed to cool at room temperature for 1 h. Further, the extracts were filtered using a strainer to remove coarse residues, and then the filtrate was filtered again using Whatman No. 1 filter paper. The pure filtrate was kept at −80 °C for 24 h and then lyophilized using a freeze drier. 

### 4.4. Source of Marine Leech Z. arugamensis

The marine leeches *Z. arugamensis* (1–1.5 cm) were procured from the aquaculture facilities. An infested hybrid grouper (*Epinephelus fuscoguttatus* × *E. lanceolatus*) (15–340 g) (diameter: 15–20 cm) with marine leeches (Figure 1A,B) was placed in a small tank containing seawater from the cage, and the leeches were removed individually by hand. The leeches were transferred into a container containing filtered seawater and incubated at 27 °C for 24 h.

### 4.5. Antiparasitic Bioassay

Adult parasites were selected, divided into five groups, and each group was provided with six parasites in a Petri dish. Group 1 served as a negative control (Figure 1C), treated with seawater only, and group 2 served as a positive control, treated with 0.25% formalin solution (Figure 1D). In contrast, groups 3, 4 and 5 were challenged with 25, 50 and 100 mg/mL of the aqueous extract of *A. indica,* respectively (Figure 1E,F). During the challenge, parasites’ inactivity and death were recorded using a stopwatch. 

### 4.6. Liquid Chromatography 

Liquid chromatography (LC) was performed using the Dionex UltiMate 3000 UHPLC system (Thermo Fisher Scientific, Waltham, MA, USA) coupled with a Thermo Syncronis C18 column (2.1 mm × 100 mm × 1.7 μm; Thermo Fisher Scientific, Waltham, MA, USA), which was maintained at 55 °C at a flow rate of 450 µL/min during analysis. The mobile phases were composed of solvent A (water added with 0.1% formic acid) and solvent B (acetonitrile added with 0.1% of formic acid). The gradient elution program was initiated at 0.5% of solvent B for 1 min, then from 0.5% to 99.5% of solvent B for 15 min and maintained for 4 min. Later, the columns were conditioned as initial for 2 min before the next injection. The injection volume was set at 2 µL.

### 4.7. Data Acquisition

MS and MS/MS data were acquired using the Thermo Scientific Q Exactive HF Orbitrap mass spectrometry system (Thermo Fisher Scientific, Waltham, MA, USA), which was equipped with a heated electrospray ionization (HESI) probe. The data acquisition was set between an *m*/*z* of 100–1000 for MS and 200–1000 for MS/MS. The resolutions of the MS and MS/MS data were acquired at 60 k and 15 k, respectively. Positive and negative HESI were both deployed at 3.5 kV and 3.0 kV, respectively. The ion source conditions were set as follows: capillary temperature of 320 °C, sheath gas flow rate of 50, aux gas flow rate of 18, sweep gas flow rate of 0 and aux gas heater temperature of 300 °C. Calibrations were performed using Pierce LTQ ESI Positive Calibrations solution and Pierce LTQ ESI Negative Calibrations solution (Thermo Fisher Scientific, Waltham, MA, USA) before sample analysis.

### 4.8. Data Analysis

The acquired data were processed and analyzed using Thermo Scientific Compound Discoverer 3.0 software (Thermo Fisher Scientific, Waltham, MA, USA) with the default settings in order to perform compound identification. Briefly, the default workflow includes background subtraction with blank data, retention time alignment, feature detection, elemental composition determination, libraries matching and fragment ion search (FISh) scoring. The identification of compounds was primarily based on the matching of MS/MS data against mzCloud and mzVault databases. Unmatched signals were attempted with the ChemSpider database using MS data and supported with a FISh scoring above 50.

### 4.9. Statistical Analysis

Data analysis was carried out using the IBM SPSS Statistics 25 Window package (IBM, Armonk, NY, USA). Significant differences between groups were determined using a one-way analysis of variance (ANOVA) followed by Tukey’s multiple comparisons test. All results were presented as the mean ±standard error of the mean (S.E.). *p* values under 0.05 were regarded as significant.

## 5. Conclusions

In the present study, it was proven that the application of various concentrations of *A. indica* aqueous extract indicated a strong antiparasitic activity with 100% mortality against the marine parasitic leech *Z. arugamensis* in an average time ranging from 6.45 to 42.65 min. The Q Exactive HF Orbitrap mass spectrometry analysis indicated the presence of five flavonoids (myricetin 3-O-galactoside, trifolin, isorhamnetin, quercetin and kaempferol), four aromatics (4-methoxy benzaldehyde, scopoletin, indole-3-acrylic acid and 2,4-quinolinediol), three phenolics (p-coumaric acid, ferulic acid and phloretin) and two terpenoids (pulegone and caryophyllene oxide). Some of the compounds were reported to have strong antiparasitic properties. This demonstrates that *A. indica* aqueous extract has the potential to act as a biocontrol agent against *Z. arugamensis* infestation. The antiparasitic properties of the aqueous extract of *A. indica* could be due to the presence of the above-mentioned metabolites. However, further investigation of the purification and isolation of the pure metabolites responsible for the antiparasitic properties is vital.

## Figures and Tables

**Figure 1 molecules-26-01908-f001:**
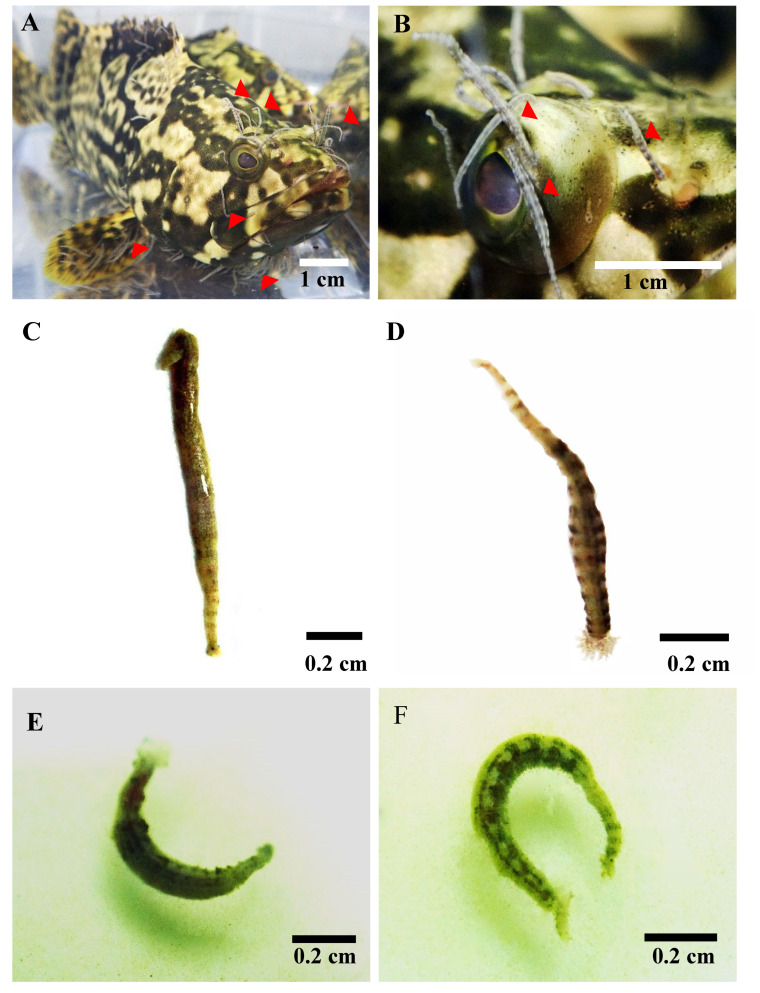
(**A**,**B**) = Hybrid grouper (*Epinephelus fuscoguttatus* ×*E.lanceolatus*), highly infested with *Z.*
*arugamensis,* indicated by the arrow (The parasites were isolated and used for the current research), (**C**) = normal parasite (exposed to seawater only, untreated with extract or chemicals, no mortality was noticed), (**D**) = formalin-exposed parasite (exposed to 0.25% of formalin solution in seawater, resulted in mortality of the parasites) (**E**,**F**) = *A. indica* aqueous extract-exposed parasites (exposed to low, medium and the higher concentration of plant extract prepared in seawater, resulted in mortality of the parasites with a wrinkled body).

**Figure 2 molecules-26-01908-f002:**
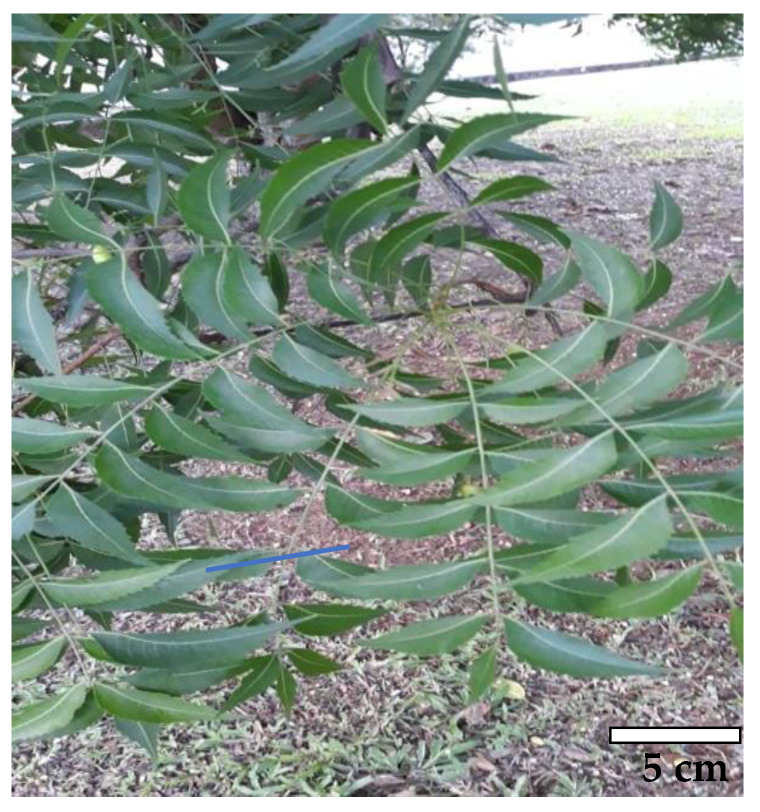
*Azadirachta indica* plant leaves.

**Table 1 molecules-26-01908-t001:** Water quality parameters of the solution for the treatment of parasitic leeches.

No.	Water Parameters		Concentrations			
	Groups	Normal Control	Positive Control (Formalin 0.25%) (*v*/*v*)	*Azadirachta indica* (mg/mL)		
				(25)	(50)	(100)
1	Temperature (°C)	24.6	25.3	24.7	24.9	25.1
2	pH	7.80	7.24	5.56	5.4	4.03
3	Salinity (ppt)	30.0	30.9	31	30.9	30.9
4	Dissolved oxygen (mg/L)	7.0	6.5	6.8	7.1	7.0

**Table 2 molecules-26-01908-t002:** Mortality time of parasitic leeches treated with different concentrations of aqueous extract of *A. indica.*

No	Group	Mortality Time (min) Mean ± S. D	Mortality (%)
1	Normal control	720.00 ± 00	0
2	Positive control (Formalin 0.25%) (*v*/*v*)	3.77 ± 0.25 ^#^	100
3	*A. indica* (25 mg/mL)	42.65 ± 9.20 ^#,$^	100
4	*A. indica* (50 mg/mL	11.69 ± 1.11 ^#,$,%,^	100
5	*A. indica* (100 mg/mL)	6.45 ± 0.45 ^#,$,%,^*	100

Each value represents the mean ± S.D. of six parasitic leeches per group. ^#^ Significance at *p* < 0.05 compared with the control group. ^$^ Significance at *p* < 0.05 compared with the formalin-treated group (0.25% *v*/*v*); ^%^ Significance at *p* < 0.05 compared with *A. indica* (25 mg/mL); * Significance at *p* < 0.05 compared with *A. indica* (50 mg/mL).

**Table 3 molecules-26-01908-t003:** Compounds identified from the aqueous extract of *A. indica* using Q Exactive HF Orbitrap mass spectrometry and Compound Discoverer 3.0. Number shows the respective compound structures can be referred in Appendix A.

No	Identified Compounds	Class	Retention Time	Formula
1	Glutamic acid	Amino acid	0.623	C_5_H_9_NO_4_
2	Arginine	Amino acid	0.687	C_6_H_14_N_4_O_2_
3	Histidine	Amino acid	0.689	C_6_H_9_N_3_O_2_
4	γ-Aminobutyric acid	Amino acid	0.734	C_4_H_9_NO_2_
5	Valine	Amino acid	0.752	C_5_H_11_NO_2_
6	Tyrosine	Amino acid	0.845	C_9_H_11_NO_3_
7	N3, N4-Dimethylarginine	Amino acid	0.854	C_8_H_18_N_4_O_2_
8	Leucine	Amino acid	0.896	C_6_H_13_NO_2_
9	Phenylalanine	Amino acid	1.202	C_9_H_11_NO_2_
10	1-Aminocyclohexanecarboxylic acid	Amino acid	1.203	C_7_H_13_NO_2_
11	4-Methoxybenzaldehyde	Aromatic	3.504	C_8_H_8_O_2_
12	Scopoletin	Aromatic	4.351	C_10_H_8_O_4_
13	2-Methylcyclohexan-1,3-dione	Cyclic ketone	1.502	C_7_H_10_O_2_
14	Jasmone	Cyclic ketone	7.041	C_11_H_16_O
15	3-Hexenoic acid	Fatty acyl	1.873	C_6_H_10_O_2_
16	9*S*,13*R*-12-Oxophytodienoic acid	Fatty acyl	7.369	C_18_H_28_O_3_
17	Decanamide	Fatty acyl	8.086	C_10_H_21_NO
18	Myricetin 3-O-galactoside	Flavonoid	4.717	C_21_H_20_O_13_
19	Trifolin	Flavonoid	5.412	C_21_H_20_O_11_
20	Isorhamnetin	Flavonoid	5.54	C_16_H_12_O_7_
21	Quercetin	Flavonoid	5.032	C_15_H_10_O_7_
22	Kaempferol	Flavonoid	5.303	C_15_H_10_O_6_
23	Pyroglutamic acid	Heterocyclic	0.827	C_5_H_7_NO_3_
24	β,β-Dimethyl-γ-methylene-γ-butyrolactone	Heterocyclic	1.612	C_7_H_10_O_2_
25	Pipecolic acid	Heterocyclic	1.049	C_6_H_11_NO_2_
26	6-Methyl-2-pyridinemethanol	Heterocyclic	1.084	C_7_H_9_NO
27	Indole-3-acrylic acid	Heterocyclic aromatic	1.935	C_11_H_9_NO_2_
28	2,4-Quinolinediol	Heterocyclic aromatic	3.241	C_9_H_7_NO_2_
29	Valylproline	Peptide	1.464	C_10_H_18_N_2_O_3_
30	Prolylleucine	Peptide	1.531	C_11_H_20_N_2_O_3_
31	p-Coumaric acid	Phenolic	3.205	C_9_H_8_O_3_
32	Ferulic acid	Phenolic	3.705	C_10_H_10_O_4_
33	Phloretin	Phenolic	5.399	C_15_H_14_O_5_
34	Guanine	Purine	0.787	C_5_H_5_N_5_O
35	Adenine	Purine	0.804	C_5_H_5_N_5_
36	1-Methyladenine	Purine	0.83	C_6_H_7_N_5_
37	2′-Deoxyadenosine	Purine	1.192	C_10_H_13_N_5_O_3_
38	2′-O-Methyladenosine	Purine	1.608	C_11_H_15_N_5_O_4_
39	Pulegone	Terpenoid	3.61	C_10_H_16_O
40	Caryophyllene oxide	Terpenoid	6.354	C_15_H_24_O
41	Nicotinic acid	Vitamin B3	0.836	C_6_H_5_NO_2_
42	Nicotinamide	Vitamin B3	0.894	C_6_H_6_N_2_O

## Data Availability

Not applicable.

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
