# Peer review of "Efficacy of the Aqueous Extract of Azadirachta indica Against the Marine Parasitic Leech and Its Phytochemical Profiling"

_molecules, 2021, doi:10.3390/molecules26071908_

Round 1

Reviewer 1 Report

The current study aimed to elucidate the anti-parasitic efficacy of the aqueous extract of the Azadirachta indica leaves against Z. arugamensis and to determine the composition using high throughput LC-Q Exactive HF orbitrap mass spectrometry. After performed these analyses, the obtained data indicated that A. indica aqueous extract could be considered a good source of metabolites with the potential to act as a biocontrol agent against the marine parasitic leech. In my opinion, this manuscript could be considered for publication in Molecules after minor revision, as follow: 1. It's not clear to me why the authors decided to investigate the aqueous extract from Azadirachta indica leaves. This plant is known to produce limonoids and other related compounds which are frequently extracted using MeOH and not H2O. Is there some specific reason to perform the extraction of plant material using H2O? 2. As described above, this plant is chemotaxonomically known as a producer of limonoids/meliacin derivatives. No related derivatives were found in the studied extract. Why? The authors recorded a 1H NMR (in high-field – 500 MHz for example) to investigate this aspect? 3. Frequently, aqueous extracts were composed of a high amount of carbohydrates. Are these compounds found in the crude studied extract? Please, include, in the revised version, a copy of the 1H NMR spectrum in order to analyze this aspect. 4. Table 3 – what “Phenolid” means? Please, revise all manuscript since several mistakes appear and must be corrected. As another example, scopoletin is coumarin and not “aromatic”. This classification is very general! 5. Figure 2 is not necessary since all identified metabolites are known compounds. As a suggestion, Figures 2 and 3 could be presented as supplementary information (SI). 6. Figure 1 appears after figure 3? Please revise – similarly to the others, figure 1 must be moved to SI.

Author Response

Reviewer # 1

Thank you very much for your comments and suggestions, we followed all your suggestions and shown in red color.

The current study aimed to elucidate the anti-parasitic efficacy of the aqueous extract of the Azadirachta indica leaves against Z. arugamensis and to determine the composition using high throughput LC-Q Exactive HF orbitrap mass spectrometry. After performed these analyses, the obtained data indicated that A. indica aqueous extract could be considered a good source of metabolites with the potential to act as a biocontrol agent against the marine parasitic leech. In my opinion, this manuscript could be considered for publication in Molecules after minor revision, as follow:

Q1: It's not clear to me why the authors decided to investigate the aqueous extract from Azadirachta indica leaves. This plant is known to produce limonoids and other related compounds which are frequently extracted using MeOH and not H2O. Is there some specific reason to perform the extraction of plant material using H2O?  

Res: We have chosen the aqueous extract of Azadirachta indica leaves due to the reason that the extract was applied to the parasitic leeches and will be applied against infested host fishes in the aquatic environment and thus, the aqueous extract of Azadirachta indica leaves will dissolve easily in that medium.

Q2: As described above, this plant is chemotaxonomically known as a producer of limonoids/meliacin derivatives. No related derivatives were found in the studied extract. Why? The authors recorded a 1H NMR (in high-field – 500 MHz for example) to investigate this aspect?

Res: In the current study, we have attempted to identify the bioactive composition possess in the aqueous crude extract using LCMS, and the range of the MS screening is limited to small molecules (<1500 m/z). Therefore, no limonoids/meliacine derivatives are detectable in the current study.

Q3: Frequently, aqueous extracts were composed of a high amount of carbohydrates. Are these compounds found in the crude studied extract? Please, include, in the revised version, a copy of the 1H NMR spectrum in order to analyze this aspect.

Res: In the current study, we have aimed to elucidate the efficacy of the aqueous extract of the Azadirachta indica leaves against Z. arugamensis and relate its efficacy to the detected bioactive compositions via LCMS analysis. We probably have detected carbohydrates, but their signals are not identifiable without chemical standards due to isomerism or chirality, thus not reported in our manuscript. Besides, the current study is aimed to indicate the presence of potential bioactive compounds, not for complete chemical profiling or proximate analysis. Detail study will be conducted to further identify and purify the bioactive compound(s) responsible for the cytotoxicity of the parasitic leech. 

Q4: Table 3 – what “Phenolid” means? Please, revise all manuscript since several mistakes appear and must be corrected. As another example, scopoletin is coumarin and not “aromatic”. This classification is very general!

Res: We apologize for the typo, it should be “phenolic”. We have checked thoroughly and corrected the spelling and grammar. We apply general classification to project the identified compounds for easy understanding. Instead of using alpha-amino acid, coumarin, indole, piperidine, and categorized the detected compounds into more classes, general classification was found easier to understand, especially for readers not from the chemistry field.

Q5: Figure 2 is not necessary since all identified metabolites are known compounds. As a suggestion, Figures 2 and 3 could be presented as supplementary information (SI). 6. Figure 1 appears after figure 3? Please revise – similarly to the others, figure 1 must be moved to SI

Res: Figures 2 and 3 have been deleted from the main text and added to supplementary information.

Reviewer 2 Report

Your paper addresses a topical scientific field, namely eco-friendly aquaculture.

The research is generally well motivated in Introduction,

In general, the research is well conducted.

The results are clearly presented and the discutions are relevant.

The bibliography is exhaustive and correctly written.

1.           In table 3: the names of some of the classes should be changed / updated, as follow:

-        currently, terms such as ,,phenolid” and ,,fatty acyl” are no longer commonly used; should be replaced (it is only a suggestion)

-        decanamide is not a fatty acyl compound; it is amide

-        scopoletin is not an aromatic compound; it is coumarin

2.           Attention to editing errors:

-        In table 3, compound [20]: Sorhamnetin (not sorhamnetin)

-        Sometimes there are double spaces before some words, for example:

R18: --Different

R22: --The

R45: --Further

R63: --The

R165: --was

R174: --(Latidae)

R188: --Encephalitozoon

R191: --[40,41]

R193: --[42,43]

R196: --end

R220: --The

R229: --The

R255: --An

R278: --Some

Author Response

Reviewer #2

Thank you very much for your comments and suggestions, we followed all your suggestions and shown in red color.

Your paper addresses a topical scientific field, namely eco-friendly aquaculture.

The research is generally well motivated in Introduction,

In general, the research is well conducted.

The results are clearly presented and the discutions are relevant.

The bibliography is exhaustive and correctly written.

Res: Thank you for your positive comments.

Q1. In table 3: the names of some of the classes should be changed / updated, as follow:

- currently, terms such as ,,phenolid” and ,,fatty acyl” are no longer commonly used; should be replaced (it is only a suggestion)

- decanamide is not a fatty acyl compound; it is amide

- scopoletin is not an aromatic compound; it is coumarin

Res: Thank you very much for your comment.

We applied the general classification to project the identified compounds for easy understanding. Instead of using alpha-amino acid, coumarin, indole, piperidine, and categorized the detected compounds into more classes, general classification was found easier to understand, especially for readers who are not from the chemistry field.

Q2:   Attention to editing errors:

- In table 3, compound [20]: Sorhamnetin (not sorhamnetin)

- Sometimes there are double spaces before some words, for example:

R18: --Different

R22: --The

R45: --Further

R63: --The

R165: --was

R174: --(Latidae)

R188: --Encephalitozoon

R191: --[40,41]

R193: --[42,43]

R196: --end

R220: --The

R229: --The

R255: --An

R278: --Some

Res: Thank you for your corrections, All corrections are followed.

In table 3, compound [20]: Sorhamnetin has been corrected

The double spaces before R18 -R278 words have been corrected.

Reviewer 3 Report

The details of my comments in the attached pdf file.

Comments:

Q1. Table1 : It is more advisable to add at the end of table the total percentage of amino acids, aromatic compounds, phenolic compounds.....etc

Q2. In the introduction part: in the paragraph started with “Azadirachta indica (neem plant) …………………. [11–13]”. You forget to report also the biological activity of plant essential oils, in fact there are various scientific research has been conducted on the antimicrobial , herbicidal and insecticidal effects of plant essential oil. Insert also the following papers in your paper.

  1. Gruľová, D., Caputo, L., Elshafie, H. S., Baranová, B., De Martino, L., Sedlák, V., Camele I. and De Feo, V. 2020. Thymol Chemotype Origanum vulgare L. Essential Oil as a Potential Selective Bio-Based Herbicide on Monocot Plant Species. Molecules, 25(3), 595. DOI: 10.3390/molecules25030595.
  2. Camele, I., Elshafie, H. S., De Feo, V. and Caputo, L. 2019. Anti-quorum Sensing and Antimicrobial Effect of Mediterranean Plant Essential Oils Against Phytopathogenic Bacteria. Frontiers in Microbiology10, 2619. DOI: 10.3389/fmicb.2019.02695.
  3. Elshafie H.S., Armentano M.F., Carmosino M., Bufo S.A., De Feo V. and Camele I. 2017. Cytotoxic Activity of Origanum Vulgare L. on Hepatocellular Carcinoma cell Line HepG2 and Evaluation of its Biological Activity. Molecules 22: (1435), 1-16. DOI: 10.3390/molecules22091435.

Q3. Results :  I think the section of  (3.2. Physiochemical parameters for the treatment of parasitic leeches) should be rearranged before (3.1. Anti-parasitic properties of the aqueous extract of A. indica)

Q4. In the discussion part: beside the information of anti-parasitic activity of the aqueous extracts of A. indica leaves, it should reported initially the particular antimicrobial activity of this plants and also other spices to demonstrated their biological activity and why you selected in this research.

Q5. In the discussion part: this part is not clear please rewrote it again (Besides, kaempferol was another widely 192 studied anti-parasitic flavonoid [42,43]. It inhibited the growth of parasite responsible for 193 amoebiasis, Entamoeba histolytica (Entamoebidae) by the alteration of cytoskeleton proteins [44].)

Author Response

Reviewer #3

Thank you very much for your comments and suggestions, we followed all your suggestions and shown in red color.

Q1. Table1: It is more advisable to add at the end of the table the total percentage of amino acids, aromatic compounds, phenolic compounds.....etc

Res: Thank you for your comment.

Both absolute and relative quantifications are not carried out in the current LCMS analysis. Besides, we only reported the putatively identified compounds, not the whole list of detected signals. We aimed to show that these compounds potentially exist in the crude, but the list is not representable for the proximate composition of the crude. This is mainly because there is only a small portion of the LCMS data is identifiable, thus only able to report those putatively identified compounds. Therefore, we cannot provide the chemical class composition of our crude extract as requested.

Q2. In the introduction part: in the paragraph started with “Azadirachta indica (neem plant) …………………. [11–13]”. You forget to report also the biological activity of plant essential oils, in fact there are various scientific research has been conducted on the antimicrobial , herbicidal and insecticidal effects of plant essential oil. Insert also the following papers in your paper.

  1. Gruľová, D., Caputo, L., Elshafie, H. S., Baranová, B., De Martino, L., Sedlák, V., Camele I. and De Feo, V. 2020. Thymol Chemotype Origanum vulgare L. Essential Oil as a Potential Selective Bio-Based Herbicide on Monocot Plant Species. Molecules, 25(3), 595. DOI: 10.3390/molecules25030595.
  2. Camele, I., Elshafie, H. S., De Feo, V. and Caputo, L. 2019. Anti-quorum Sensing and Antimicrobial Effect of Mediterranean Plant Essential Oils Against Phytopathogenic Bacteria. Frontiers in Microbiology, 10, 2619. DOI: 10.3389/fmicb.2019.02695.
  3. Elshafie H.S., Armentano M.F., Carmosino M., Bufo S.A., De Feo V. and Camele I. 2017. Cytotoxic Activity of Origanum Vulgare L. on Hepatocellular Carcinoma cell Line HepG2 and Evaluation of its Biological Activity. Molecules 22: (1435), 1-16. DOI: 10.3390/molecules22091435.

Res: Thank you for your comment

In the introduction part we have added the information regarding the biological activity of the essential oil of A. indica and also the requested papers have been added in the text as

‘’Extracts and essential oils from of a natural product can act as herbicide, antimicrobial, anticancer and anti-parasitic agent [11–14].

The essential oils of A. indica have been reported with antimicrobial property against Enterococcus faecalis, Aerococcus viridans,  Pseudomonas aeruginosa, Proteus mirabilis and Escherichia coli and antiparasitic activity against caligid parasites on seabass[20,21]”

Q3. Results: I think the section of (3.2. Physiochemical parameters for the treatment of parasitic leeches) should be rearranged before (3.1. Anti-parasitic properties of the aqueous extract of A. indica)

Res: Thank you. The section Physiochemical parameters for the treatment of parasitic leeches have been rearranged before  the section of Anti-parasitic properties of the aqueous extract of A. indica

Q4. In the discussion part: beside the information of anti-parasitic activity of the aqueous extracts of A. indica leaves, it should reported initially the particular antimicrobial activity of this plants and also other spices to demonstrated their biological activity and why you selected in this research.

Res: The information regarding the biological activity and the reason for the selection of the A.india has been mentioned in the discussion part as

 “In this study, we selected A. india due to its antimicrobial, anti-inflammatory, antipyretic, insecticidal and acaricidal nature [15–17,22] and determined the anti-parasitic potential of the aqueous extract of A. indica leaves.”

Q5. In the discussion part: this part is not clear please rewrote it again (Besides, kaempferol was another widely studied anti-parasitic flavonoid [42,43]. It inhibited the growth of

parasite responsible for  amoebiasis, Entamoeba histolytica (Entamoebidae) by the alteration of cytoskeleton proteins [44].)

Res: Thank you for your comment

Now, it is rephrased in the discussion part as: “Kaempferol inhibited the growth of parasite Entamoeba histolytica (Entamoebidae) responsible for amoebiasis by altering cytoskeleton proteins [47]”

Q6: Line 139, Table 3. Tentatively identified compounds from aqueous extract of A. indica using Q Exactive HF 140 Orbitrap mass spectrometry and Compound title should rewritten.

Res:  Thank you  for your comments, Yes the title has been changed as

Table 3.  Compounds identified from aqueous extract of A. indica using Q Exactive HF Orbitrap mass spectrometry and Compound Discoverer 3.0.

Q7: Line 140.  At the end of this table you must write the total percentage of amino acids, aromatic compounds,.....etc

Res: Both absolute and relative quantifications are not carried out in the current LCMS analysis. Besides, we only reported the putatively identified compounds, not the whole list of detected signals. We aimed to show that these compounds potentially exist in the crude, but the list is not representable for the proximate composition of the crude. This is mainly because there is only a small portion of the LCMS data is identifiable, thus only able to report those putatively identified compounds. Therefore, we cannot provide the chemical class composition of our crude extract as requested.

Q8: line 150, The scientific names should be written in the first mention then you can write the name of the plant directly or abbreviation.

Res: Thank you for your comments

The correction has been done Z. arugamensis  has been changed to “Zeylanicobdella arugamensis

Q9: Line 182, write only the abbreviation of the instrument

Res: Some of the mass spectrometry names have been shortened to "Orbitrap mass spectrometry".

Q10: line 225, do not write the symbol abbreviation in the titles or subtitles.

Res: Yes the abbreviation LC has been deleted.
